# Morphological, Pathogenic and Toxigenic Variability in the Rice Sheath Rot Pathogen *Sarocladium Oryzae*

**DOI:** 10.3390/toxins12020109

**Published:** 2020-02-08

**Authors:** Kaat José Peeters, Ashley Haeck, Lies Harinck, Oluwatoyin Oluwakemi Afolabi, Kristof Demeestere, Kris Audenaert, Monica Höfte

**Affiliations:** 1Laboratory of Phytopathology, Department of Plants and Crops, Faculty of Bioscience Engineering, Ghent University, Coupure Links 653, 9000 Ghent, Belgium; kaat.peeters@ugent.be (K.J.P.); afolabi_toy@yahoo.com (O.O.A.); 2Research Group EnVOC, Department of Green Chemistry and Technology, Ghent University, Coupure Links 653, 9000 Ghent, Belgium; ashhaeck@hotmail.com (A.H.); lies.harinck@ugent.be (L.H.); kristof.demeestere@ugent.be (K.D.); 3Laboratory of Applied Mycology and Phenomics, Department of Plants and Crops, Faculty of Bioscience Engineering, Ghent University, Valentin Vaerwyckweg 1, 9000 Ghent, Belgium; kris.audenaert@ugent.be

**Keywords:** *Oryza sativa*, cerulenin, helvolic acid, sectorisation, phenotypic stability, LC–MS/MS, Africa

## Abstract

Sheath rot is an emerging rice disease that leads to considerable yield losses. The main causal agent is the fungus *Sarocladium oryzae.* This pathogen is known to produce the toxins cerulenin and helvolic acid, but their role in pathogenicity has not been clearly established. *S. oryzea* isolates from different rice-producing regions can be grouped into three phylogenetic lineages. When grown in vitro, isolates from these lineages differed in growth rate, colour and in the ability to form sectors. A diverse selection of isolates from Rwanda and Nigeria, representing these lineages, were used to further study their pathogenicity and toxin production. Liquid chromatography high-resolution mass spectrometry analysis was used to measure cerulenin and helvolic acid production in vitro and in planta. The three lineages clearly differed in pathogenicity on the *japonica* cultivar Kitaake. Isolates from the least pathogenic lineage produced the highest levels of cerulenin in vitro. Helvolic acid production was not correlated with the lineage. Sectorisation was observed in isolates from the two least pathogenic lineages and resulted in a loss of helvolic acid production. In planta, only the production of helvolic acid, but not of cerulenin, correlated strongly with disease severity. The most pathogenic isolates all belonged to one lineage. They were phenotypically stable, shown by the lack of sectorisation, and therefore maintained high helvolic acid production in planta.

## 1. Introduction

Rice is the main staple food for more than half of the world’s population [1,2]. More than 500 million tons of milled rice were produced in 2017, of which 30,000 tons were from Africa [3]. Due to population growth, urbanisation and dietary changes, rice consumption in Africa grows at about 5.5% per year (2000–2010 average) [1,4]. As the demand for rice strongly exceeds the local production, 43% of the rice consumed in Africa is imported, which costs more than USD 1.5 billion per year [5,6]. There is, however, great potential to increase the rice yield in Africa by improving crop management and expanding rice production. The major constraints are abiotic stresses (drought or excess of water, nutrient deficiencies and extreme temperatures) and biotic stresses (weeds, diseases and pests) [4].

Sheath rot is considered one of the most important emerging diseases of rice, causing yield losses from 20–30% up to 85% [7,8,9]. This disease is mainly caused by the seed-borne fungus *Sarocladium oryzae* [10]. Since its first description in 1922 in Taiwan as *Acrocylindrium oryzae*, the fungus has spread worldwide. Sheath rot occurs in both rainfed and irrigated ecosystems, affecting all rice varieties. Dwarf and high-yield Asian varieties are the most susceptible. *S. oryzae* infection becomes visible on the flag leaf sheath as greyish-brown necrotic lesions, enlarging as the disease progresses until the whole leaf sheath is necrotic. Enclosed panicles are affected, leading to sterile, empty or discoloured grains or, during severe infection, only partial or no emergence of the panicle [9,10]. *S. oryzae* can survive in seeds, plant residues, weeds and soil and it is transmitted by seeds, wind and insects. Wounds and stomata are needed to enter the host [7,10].

Cell wall degrading enzymes and the phytotoxins helvolic acid, cerulenin and SO-toxin are described to be the main pathogenicity factors used by *S. oryzae* [11,12]. Clear evidence for the role and importance of cerulenin and helvolic acid in the disease development of sheath rot, however, is still missing. Helvolic acid is a tetracyclic triterpenoid that is able to bind magnesium-ions, thereby interfering with chlorophyll biosynthesis, which leads to chlorosis in the host tissue [13,14]. This compound is structurally very similar to the well-known steroid antibiotic fusidic acid, produced by *Fusidium coecineum*. Both compounds are toxic to Gram-positive bacteria [15,16]. Helvolic acid is produced by various fungi including the entomopathogenic fungus *Metarhizium anisopliae* and the opportunistic human pathogen *Aspergillus fumigatus* [17]. Also, the hexaketide amide, cerulenin, is both phytotoxic and antimicrobial. This compound interferes with fatty acid and polyketide biosynthesis by inhibiting the malonyl-ACP:acyl-ACP condensation step [18]. Its antifungal activity against different fungi, among which is *Sclerotium oryzae* and *Magnaporthe oryzae*, is well described [14,19,20]. The production of secondary metabolites, with both phytotoxic and antimicrobial actions, improves the competitive ability of *S. oryzae*. Thereby its survival increases and, consequently, its pathogenic potential increases as well [14].

Secondary metabolite production is known to be favoured by suboptimal growth conditions, which reflect the natural environment. Here, fungi are subjected to different environmental stimuli such as carbon sources, reactive oxygen species and interspecies communication. These stimuli can induce secondary metabolism [21,22]. When, however, the level of stress exceeds the tolerance level of the fungus, culture degeneration can occur. In culture, this process can lead to the formation of morphological variants that are called sectorisation. Sectors can vary in their pigmentation and are often impaired in their ability to produce secondary metabolites, pathogenicity-related enzymes and spores [23,24,25,26,27,28]. Since the frequency of sector formation is negatively correlated with the stability of the isolate in the natural environment and its virulence, this process should be taken into account when studying pathogen populations [23,26,28].

The occurrence of rice sheath rot can cause considerable yield losses, and effective control measures are not available yet. Because of the different causal agents of sheath rot, and the lack of knowledge about their mode of action and resistance factors, there are no resistant varieties [8,12]. Seed treatments with fungicides are not effective because they do not kill the fungi inside the glume [7,12]. *Pseudomonas fluorescens* has been studied as a possible biocontrol agent against *S. oryzae* but the data in the field are still inconsistent [7,29]. In order to find effective control measures against this disease, it is important to elucidate the mode of action of the causal agent and study the diversity of the *S. oryzae* population.

Previous diversity studies in India and Bangladesh [12,30] have described the morphological and genetic variations and the pathogenicity of the *S. oryzae* population in different agro-ecological regions. A study that includes isolates from different regions worldwide, however, is still missing. Also, the population of *S. oryzae* in Africa is poorly studied and the etiology of rice sheath rot still needs to be further investigated. Both cerulenin and helvolic acid production have been measured in in vitro cultures but, so far, the toxin production in the host plant has not yet been quantified [13,14].

Based on the partial sequences of the internal transcribed spacer and actin genes, *S. oryzae* can be divided into three lineages [31,32,33]. In this work, we studied these phylogenic groups, based on *S. oryzae* isolates from Rwanda and Nigeria together with *S. oryzae* isolates originating from different rice-producing countries worldwide. The morphological and pathogenic variability was studied together with the in vitro and in planta toxin production of these isolates. By studying the production of helvolic acid and cerulenin during the infection process of isolates with a different level of pathogenicity, we aim to elucidate the importance of these toxins in the infection process of the host.

## 2. Results

### 2.1. Morphological Characterisation and Growth Rate

Isolates from the three phylogenetic lineages of *S. oryzae* were grown on potato dextrose agar (PDA) and oatmeal agar (OA) for 15 days to study their morphology and growth rate. The radial growth rate was calculated based on the colony diameter at five time points (Figure 1). With an average radial growth rate of 1.1 mm.day^−1^, isolates of Group 1 grew, except for CBS120.817 and CBS485.80, significantly slower on PDA than the isolates of Groups 2 and 3. The radial growth rate varied a lot (from 1.3 to 2.5 mm.day^−1^) among the isolates of Group 2, while Group 3 showed a more constant radial growth rate of meanly 1.8 mm.day^−1^ (Figure 1A,C). On oatmeal agar (OA) the radial growth rate was more constant than on PDA (Figure 1A,B). When grown on OA, all three groups had a mean radial growth rate between 1.9 and 2.0 mm.day^−1^ (Figure 1C). Although the variation among the isolates of one group was higher than this difference of 0.1 mm.day^−1^, the variation was still smaller on OA (Figure 1B).

As shown in Figure 2, the colony colour of the *S. oryzae* isolates, grown on PDA, varied between white, pale salmon and orange. The brightest coloured isolates belonged to Group 1 (RFRG2 and IBNG0013). The colonies grew powdery and were radially folded, pale to dark orange with a white border. Isolates of Group 2 had radially folded (RFNG33) or radially striated (RFBG9) white or pale salmon colonies. The mycelium growth varied from powdery to wet aerial. Colonies of Group 3 were morphologically the most uniform and the most distinct from the other groups. All isolates formed powdery, dark or pale orange colonies on PDA, and some isolates produced a deep yellow diffusible pigment (BDNG0025). Isolates of Groups 1 and 2 showed sectorisation on PDA. For the isolates of Group 3, no sector formation was observed. The sectors differed in their morphological characteristics such as growth rate and colour.

The Group 1 isolate CBS180.74 was used to further study sectorisation. Four different phenotypes were observed on PDA. Being the original and most abundant phenotype, the white phenotype was considered as the parent culture. This white phenotype showed a radially folded, powdery growth with a light salmon-green shade. On PDA, sectors formed in this culture in three different phenotypes. The phenotypes mainly differed in colour (brown, green, orange) and growth rate. Some phenotypes grew more powdery than the parent and all cultures were radially folded (Figure 2).

### 2.2. Pathogenicity

To evaluate the pathogenicity of *S. oryzae*, five isolates representing each of the three phylogenic groups were used in pathogenicity experiments. Plants of the *japonica* cultivar Kitaake were inoculated with all fifteen isolates. The disease development was evaluated at 8 days post-inoculation (DPI) by measuring the lesion area. The variation in virulence among the isolates of one group is high. For example, CBS414.81 caused a mean lesion area of 30 mm^2^, while BDNG0025 caused a mean lesion area of 278 mm^2^ (Figure 3A). The mean lesion area on Kitaake caused by Group 2 was significantly higher than the mean area caused by Group 1. Overall, Group 3 showed significantly higher pathogenicity than Groups 1 and 2, but all *S. oryzae* isolates tested were able to induce sheath rot symptoms on Kitaake (Figure 3).

### 2.3. In Vitro Cerulenin and Helvolic Acid Production

To evaluate the production of cerulenin and helvolic acid by *S. oryzae*, sixteen isolates belonging to the three phylogenic groups were used in in vitro experiments. For most Group 1 and 3 isolates, cerulenin levels were 10–20-fold higher than helvolic acid levels (Figure 4A,B). For Group 1 and 2 isolates, the in vitro production of cerulenin showed a very strong positive correlation with the in vitro production of helvolic acid (respectively, r = 0.699, *p* < 0.001, *n* = 54 and r = 0.908, *p* < 0.001, *n* = 54). A weak, negative, non-significant correlation was observed for the in vitro production of both toxins by isolates of Group 3 (r_s_ = −0.241, *p* = 0.246, *n* = 54).

With a mean cerulenin production of 6922 µg L^−1^, Group 1 contained the highest producers. Group 2 contained the lowest producers (on average 97 µg L^−1^) and isolates of Group 3 produced moderate levels of cerulenin (on average 1409 µg L^−1^). Despite the clear difference in the mean cerulenin production of the three phylogenic groups, there was a large variation in production among the isolates of one group. Only two isolates of Group 2, RFNG41 and RFNG33, produced cerulenin while no cerulenin could be detected in the PDA cultures of the other four isolates. Also Group 3 contained isolates producing cerulenin from 117 (BDNG0025) up to 2914 µg L^−1^ (IBNG0008) (Figure 4A). Although there was also a lot of variation in the helvolic acid production of the *S. oryzae* isolates, the differences among the three groups were smaller than in the cerulenin production. Group 1 isolates were the highest producers of helvolic acid with concentrations around 450 µg L^−1^ except for the Rwandan isolate RFRG2, which produced no detectable levels of helvolic acid (Figure 4B). Helvolic acid levels of Group 2 varied from no detection to 348 µg L^−1^. All isolates of Group 3, on the other hand, produced about 300 µg L^−1^ of helvolic acid (Figure 4B).

### 2.4. Cerulenin and Helvolic Acid Production by Different Phenotypes of CBS180.74

The conservation of the toxin production after sectorisation of CBS180.74 was studied. Therefore, all four phenotypes were grown in a liquid culture of which the extract was analysed by liquid chromatography high-resolution mass spectrometry. The results showed that the production of both cerulenin and helvolic acid was affected by sectorisation. The parent culture (white) produced the highest levels of both toxins (Figure 5). The green phenotype was most affected in its cerulenin production, with a decrease of almost 50%. Compared to the parent culture, all phenotypes showed a very strong decrease in their helvolic acid production with no detection of helvolic acid in the orange culture at all (Figure 5).

### 2.5. A Comparison of the Toxin Production in Vitro and in Planta

Ten *S. oryzae* isolates representing the three phylogenic groups were used to compare toxin production in vitro and in the host plant. Starting from the same culture on PDA, in vitro cultures for toxin measurements and grain inoculum for rice inoculation were set up. Cerulenin and helvolic acid levels were measured in 7-day-old in vitro cultures and the rice sheath of plants at 6 DPI. At day 7, the high cerulenin-producing isolates IBNG0008, IBNG0009, RFRG2, CBS180.74 and BDNG0005 produced about ten times more cerulenin than helvolic acid on PDA (Figure 6A,B). In the rice sheath however, at 6 DPI, the measured concentration of helvolic acid was higher than the cerulenin concentration except for RFRG2, for which no helvolic acid production could be detected in any condition (Figure 6C,D).

The in vitro production of cerulenin did not associate with its production in the host (Figure 7F). Nevertheless, the high cerulenin producers, such as IBNG0008, IBNG0009, RFRG2, CBS180.74 and BDNG0005, produced in both conditions significantly more cerulenin than the medium producers, BDNG0025 and RFNG41. Additionally, for isolates that did not produce any cerulenin on PDA (RFNG122, RFNG30 and RFBG3), no or very low levels (on average 1 ng/g, RFNG122) of cerulenin could be detected in the rice sheath (Figure 6A,C).

The in vitro production of helvolic acid did not correlate with the production in planta although the high in vitro-producing isolates (IBNG0008, IBNG0009, BDNG0025, CBS180.74, RFNG41 and BDNG0005) produced high to moderate amounts in the rice plant (Figure 7E). The production profiles of the low producing isolates, RFNG122 and RFBG3, were similar in both conditions and RFRG2 produced no helvolic acid in culture, nor in the rice sheath (Figure 6A–D).

### 2.6. Disease Severity Correlates with In Planta Helvolic Acid Production

In order to correlate toxin levels with the severity of the caused symptoms, disease development was evaluated by measuring the lesion area (Figure 6E). In vitro production of cerulenin or helvolic acid was not correlated with pathogenicity (Figure 7A,B). Cerulenin levels in the rice sheath did not associate with the lesion area either (Figure 7D). The production of helvolic acid in the rice sheath, however, showed a very strong, positive, linear correlation with the disease severity (r = 0.898, *p* < 0.001, *n* = 10; Figure 7C).

## 3. Discussion

In this study, *S. oryzae* isolates from an international collection were used to describe the morphological, toxigenic and pathogenic variations. The population of *S. oryzae* can be divided into three distinct lineages [31]. Concerning the high level of conservation of housekeeping genes, this distinction suggests that the population of *S. oryzae* is very diverse. This observation was confirmed by the phenotypic and pathogenic characterisation of the isolates used in this study. Especially when grown on PDA, the isolates belonging to Groups 1 and 2 showed a variable growth rate and colour pattern. Isolates from Group 3 were more uniform in growth rate and colour than isolates from Groups 1 and 2. In addition, isolates from Groups 1 and 2 were less pathogenic than isolates belonging to Group 3.

To investigate these differences in pathogenicity between the lineages and among the isolates of the groups, the production of the phytotoxins cerulenin and helvolic acid in PDA cultures was compared with the production in host tissue. Cerulenin production was highest for Group 1 isolates and Group 2 isolates produced no or very low levels of cerulenin in vitro. Helvolic acid production, on the other hand, was not correlated with the phylogenetic groups. The observed levels of both toxins in culture were in agreement with the levels measured by Ayyadurai et al. (2005) [12]. Up till now, these toxins were thought to be the main pathogenicity factors of *S. oryzae* [34]. Isolates that produce high amounts of helvolic acid and/or cerulenin in culture were described to cause a higher disease incidence [12,14]. In this work, however, these observations could not be confirmed since the production of cerulenin or helvolic acid by an isolate in vitro did not correlate with its pathogenicity.

To the best of our knowledge, neither cerulenin nor helvolic acid has previously been measured in the plant. In this work, a new LC–MS/MS method was established to reliably quantify both toxins in the rice sheath. Interestingly, the production of cerulenin and helvolic acid in vitro did not correlate with the production in the rice sheath. Nor did the cerulenin levels in the rice sheath associate with pathogenicity. A strong positive correlation, however, was observed between pathogenicity and in planta helvolic acid levels, with the highest in planta helvolic acid producers all belonging to Group 3. The fact that Group 3 isolates are able to produce high levels of helvolic acid in the rice sheath, and therefore are more virulent than the isolates of the other two groups, could possibly be attributed to a lack of sectorisation. Isolates of Group 3 were phenotypically stable in culture, while most isolates of Groups 1 and 2 formed morphological variants, called sectors. Sectorisation can be the result of genetic changes, such as spontaneous mutations or transposons, of cultural degeneration caused by ageing or stress conditions, epigenetic changes or mycovirus infection [23,26,28,35]. Sectors are often impaired in their ability to produce secondary metabolites [23,36]. This was confirmed in this study for the Group 1 isolate CBS180.74. Helvolic acid levels were much lower in the liquid cultures of all morphological variants compared to the parent culture. Also, the cerulenin production by the green and orange variants was lower than the production by the parent culture. Due to this decrease in toxin production, unstable isolates are often less virulent [23,26,35].

Shah and Butt (2005) [26] describe that more sectors are formed in nutrient-rich medium. As the growth of *S. oryzae* isolates was more stable on OA medium than on PDA, our study confirms these observations. In PDA, high nutrient levels could cause osmotic stress, leading to sectorisation [37]. In planta, *S. oryzae* encounters multiple stresses, including oxidative stress [35]. During the invasion of the host, a pathogen is subjected to high levels of reactive oxygen species (ROS) produced by both the pathogen itself and by the host, which leads to phenotypic degradation [38]. The lack of sector formation by isolates of Group 3 could possibly indicate that this lineage is more tolerant to stress.

Another explanation for the observed differences in stability and virulence between Group 3 and the Group 1 and 2 isolates could be a viral infection of the latter. Mycovirus infection is a well-described cause of sectorisation in plant pathogens, leading to hypovirulence [25,36]. The spread of these mycoviruses occurs mainly through hyphal anastomosis. This process of horizontal gene transmission occurs between vegetative compatible isolates [24,36,39]. Since we only observe sectorisation and hypovirulence in the isolates of two groups (Groups 1 and 2), vegetative incompatibility with Group 3 could possibly explain the high virulence and stability of the Group 3 isolates [31,32]. The difference in toxin production between the in vitro conditions and the rice sheath could also indicate that the isolates of Groups 1 and 2 are indeed infected with a mycovirus [39]. Brusini and Robin (2013) [40] observed that the transmission rate of mycoviruses is higher in the host plant of the pathogenic fungus, leading to more symptoms of the viral infection in planta than in vitro. Both phenotypic degradation and mycovirus infection can lead to a decreased production of the phytotoxins, resulting in lower pathogenicity. In accordance with this hypothesis, we observed a decrease in the relative toxin production of the isolates of Groups 1 and 2 in the host tissue compared to the production on PDA.

Based on these results, we conclude that in planta toxin measurements are more relevant to investigate the role of the phytotoxins in the etiology of sheath rot. As the levels of helvolic acid in infected tissue strongly correlate with the pathogenicity, our data suggest that helvolic acid is an important pathogenicity factor of *S. oryzae*. We also conclude that not only the competence to produce helvolic acid is an important trait, but also the ability to maintain this production in stressed conditions. *S. oryzae* is also known for its production of cell wall degrading enzymes [12]. As the production of pathogenicity-related enzymes may be affected during sectorisation, a decrease in this enzyme activity could possibly explain the low levels of pathogenicity of the high helvolic acid-producing isolate BDNG0005, which clusters in Group 1 [26].

To further investigate this hypothesis, future research should study the presence and transmission of mycoviruses in the different phylogenetic groups of *S. oryzae*. Also, stress resistance and its relation to pathogenicity and toxin production of the different groups should be studied.

## 4. Materials and Methods

### 4.1. Fungal Isolates

In this study, isolates from traditional rice-growing areas in Rwanda (Bugarama, Nyagatare and Rwagitima, and Rugeramigozi) and Nigeria (Ibadan and Badaggi) were used together with isolates from the CBS collection (Table A1) [31,32,41]. Pure cultures were stored on PDA (Difco) slants and at −80 °C in 20% glycerol.

### 4.2. Morphological Characterization

To measure growth, isolates were grown on 50% OA (Difco) and PDA plates at 28 °C. During 15 days, colony diameters were measured every 3 days. For each plate, two perpendicular diameters were measured, of which the average was used for the radial growth rate calculations. The growth rate was calculated by performing a linear regression on the average diameter at five different time points. For each isolate, the radial growth rate was measured three times on each medium.

### 4.3. Pathogenicity Assays

Fifteen isolates representing all three groups were selected for pathogenicity tests on rice plants. Isolates from Rwanda (regions Bugarama, Nyagatare and Rwagitima, and Rugeramigozi), Nigeria (regions Ibadan and Badaggi) and the CBS collection were included in the pathogenicity test.

Inoculum was prepared according to the standard grain inoculum technique [42]. Briefly, rice grains were soaked in water for 60 min, excess water was removed, and the grains were autoclaved twice on two different days.

For each 4 g of rice grains, one plug (diameter = 5 mm) from the edge of a 14-day-old fungal colony was added together with 1 mL of sterile distilled water. Every 2 days, the grain inoculum was shaken to prevent the formation of clumps. After 14 days of incubation at 28 °C, the inoculum was fully colonised.

Rice seeds of the *japonica* cultivar Kitaake were dehulled and surface sterilised in 2% sodium hypochlorite solution for 25 min, rinsed five times in sterile distilled water, and placed in Petri dishes containing sterile moistened filter papers (Whatman, grade 3). After 1 week of incubation at 28 °C, seedlings were sown in perforated plastic trays (22 × 15 × 6 cm) containing sterile potting soil (Structural; Snebbout, Kaprijke, Belgium). Rice seedlings were maintained in a greenhouse (28 °C, 60% relative humidity) and fertilized weekly with 0.2% iron sulphate and 0.1% ammonium sulphate.

One fully colonised grain was introduced in the junction point between the sheath of the second youngest plant leaf and the stem. Inoculation points were covered with moist cotton and wrapped with parafilm to maintain humidity. Plants were incubated under growth chamber conditions (28 °C day/28 °C night, 12/12 light regimen, and 85% relative humidity during the first 24 hrs, 65% relative humidity during day 2–8). The disease development was evaluated 8 days after inoculation by measuring the lesion area (π × (lesion length/2) × (lesion width/2)) on the flag leaf sheath. For each taxonomic group, the experiments included five isolates with distinct morphologic characteristics. Each treatment consisted of three trays containing five plants.

### 4.4. Chemical Analysis

#### 4.4.1. Extraction from PDA

The toxin production in solid medium was measured for all fifteen isolates used in the pathogenicity experiments and the Group 2 isolate, RFNG41. Each isolate was grown in one 6-well plate with each well containing 3 mL of PDA. Five wells were inoculated with one plug of a 2-week-old fungal colony, and one non-inoculated well was used as a control for contamination. After a seven-day incubation period at 28 °C in constant dark, the total content of each well was cut in small pieces and toxins were extracted in an hour with 3 mL of chloroform [12,43]. Agar and mycelium were removed by filtration using sterile filter paper (Whatmann, grade 3). The collected chloroform phase was reduced to dryness under N_2_ at 20 °C in a Turbovap^®^ LV automated concentration evaporator (Biotage, Uppsala, Sweden). After reconstituting the extract with methanol/water (20:80 *v/v*) containing 0.1% formic acid, samples were ready for instrumental analysis.

#### 4.4.2. Extraction from Liquid Culture

For the extraction from a liquid culture, *S. oryzae* isolates were grown for 7 days at 28 °C in total dark in still liquid culture in test tubes. Every isolate was grown in five test tubes, each containing 10 mL liquid medium and three plugs (diameter 5 mm) of a 2-week-old fungal colony grown on PDA. The liquid medium contained 1% glucose, 3% glycerol, 0.5% peptone and 0.2% NaCl [44]. The 7-day-old liquid culture was filtered using sterile filter paper (Whatman, grade 3) to remove the mycelium after which the filtrate was filter-sterilized with a 0.45 µm syringe filter unit (Millex). An equal amount of chloroform was added to the sterile filtrate and after shaking for 10 s, the filtrate was incubated for 1 h in the dark [12]. After incubation, the water phase was collected and discarded, leaving a pure chloroform extract. Further steps are described in Section 4.4.1.

#### 4.4.3. Extraction from Plant Material

The plants were inoculated with ten *S. oryzae* isolates belonging to the three phylogenic groups using the standard grain inoculum technique as described above. At 6 DPI, samples were collected by pooling the inoculated area of the sheath of all five plants of one tray. The samples were immediately immersed into liquid N_2_ and stored at −80 °C. Just before the extraction, the material was ground using a tissuelyser. Of this plant powder, 100 mg was extracted by adding 5 mL cold (approximately 4 °C) modified Bieleski solvent (methanol, ultrapure water and formic acid, 75:20:5) and vortexing for 10 sec. After 20–24 hrs of incubation at −80 °C, 4 mL of the supernatant was filtered (30 kDa Amicon Ultra centrifugal filter, Merck Millipore, Overijse, Belgium) for 30 min at 3,900 rpm at 4 °C in a SW9 R centrifuge (Froilabo, Paris, France). Next, 2.5 mL of the filtrate was reduced to dryness under N_2_ at 20 °C using a Turbovap^®^ evaporator. When the extract was reconstituted with 0.5 mL methanol/water (20:80 *v/v*) with 0.1% formic acid, it was ready for instrumental analysis.

#### 4.4.4. Instrumental Analysis

Instrumental analysis was performed on an ultra-high performance liquid chromatography system coupled to a quadrupole-orbitrap mass spectrometer, as described in detail by Haeck et al. (2018) [45]. Chromatographic separation was achieved on an Accela U-HPLC pumping system (Thermo Fisher Scientific, Erembodegem, Belgium), coupled to an Accela autosampler and degasser and equipped with a Nucleodur C18 column (50 × 2 mm; 1.8 µm d_p_, Macherey-Nagel, Düren, Germany). The mobile phase consisted of (A) methanol with 0.01% formic acid and (B) water with 0.1% formic acid, and the linear gradient was (for solvent A): 0–1 min at 20%, 1–2.5 min from 20–45%; 2.5–9 min from 45–100%; 9–10 min at 100%; 10–14 min at 20%. Mass spectrometric analysis was carried out using a Q-Exactive™ bench top HRMS (Thermo Fisher Scientific Erembodegem, Belgium), equipped with a heated electrospray ionization source operating in the positive ionisation mode. The measurements were performed in targeted single ion monitoring (t-SIM) at a mass resolving power of 70,000 full width at half maximum (FWHM). The method has been validated according to Haeck et al. (2018) [45] by the use of analytical standards (>96% purity) purchased at Abcam (Cambridge, UK). The analytical process efficiency, including both the extraction recovery and the matrix effects, ranged between 70% (helvolic acid) and 114% (cerulenin). Interday repeatability data showed good precision, with relative standard deviations (RSD, *n* = 3) lower than 10% at concentration levels close to the analyte’s limits of detection (0.05 and 0.08 µg.L^−1^ for helvolic acid and cerulenin, respectively). The linearity of the instrumental analysis is expressed by a R^2^ > 0.965 (*n* = 8) within at least three orders of magnitude.

### 4.5. Statistical Analysis

All statistical tests were performed with SPSS 25.0 (IBM SPSS, Armonk, NY, USA) with a significance level fixed at 0.05. To test the assumption of normality, the Shapiro–Wilk test (Bonferroni correction) was used. Homoscedasticity was tested using Levene’s test. If both assumptions were fulfilled, a one-way ANOVA (post-hoc: Tukey–Bonferroni) test was used to compare the means. In case at least one of these assumptions was not met, the non-parametric Kruskal–Wallis rank sum test was performed, followed by the Mann–Whitney test for a pairwise comparison. To test for correlations, the Pearson correlation (r) was used. If the relationship was non-linear, Spearman’s rank (r_s_) correlation was applied.

## Figures and Tables

**Figure 1 toxins-12-00109-f001:**
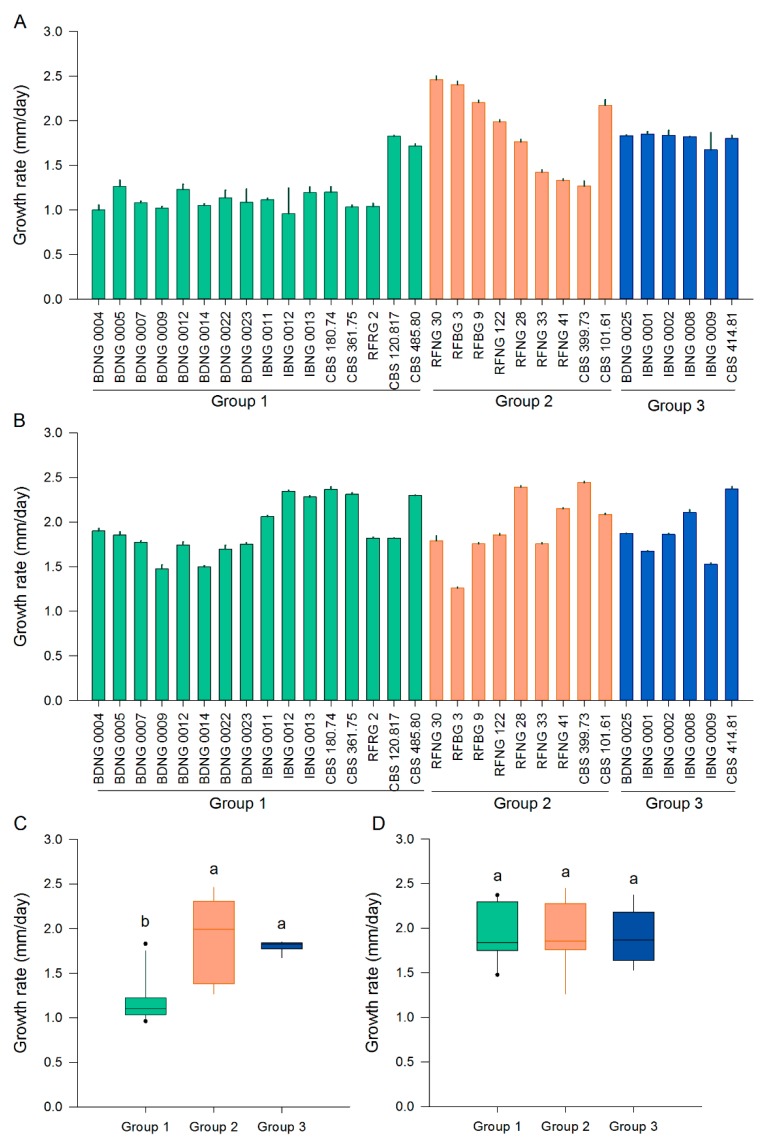
Radial growth rate of *S. oryzae* isolates grown on potato dextrose agar (PDA) or oatmeal agar (OA). (**A**) Radial growth rate of individual isolates on PDA. (**B**) Radial growth rate of individual isolates on OA. All bars show the mean ± SE with three replicates for each isolate. (**C**) Radial growth rate of all isolates belonging to Groups 1, 2 and 3 on PDA. (**D**) Radial growth rate of all isolates belonging to Groups 1, 2 and 3 on OA. Isolates of Groups 1, 2 and 3 are represented by, respectively, green, orange and blue boxplots. Boxplots represent the median with the first and third quartile, the whiskers show the minimum and maximum values. Outliers and extreme values are represented by dots. Boxplots marked with different letters are statistically different (PDA: Mann–Whitney, 18 ≤ *n* ≤ 48, α = 0.05; OA: ANOVA, 18 ≤ *n* ≤ 48, α = 0.05).

**Figure 2 toxins-12-00109-f002:**
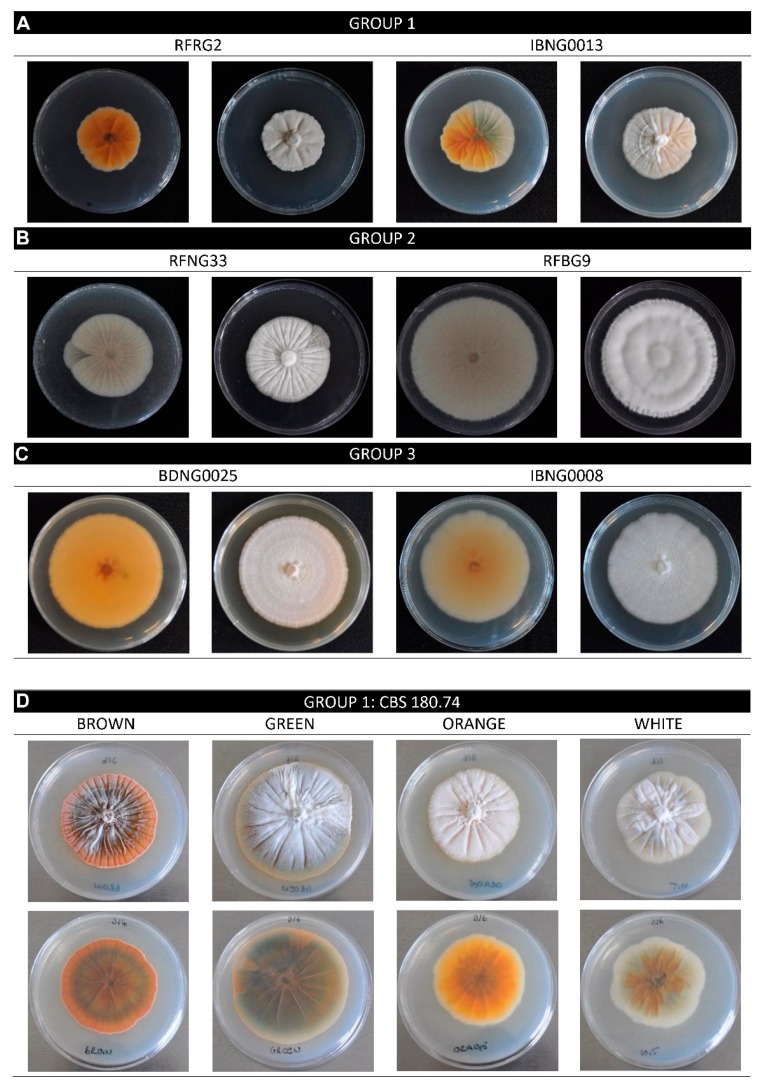
*S. oryzae* shown on PDA. **(A-C**) Isolates of Group 1, 2 and 3 when grown for 15 days in dark at 28 °C. Left is reverse view, right is front view. Sectorisation is clearly visible for IBNG0013 (Group 1) and RFNG33 (Group 2). (**D**) Isolated sectors of the Group 1 strain CBS180.74, grown for 21 days in dark at 28 °C. Top is front view, bottom is reverse view.

**Figure 3 toxins-12-00109-f003:**
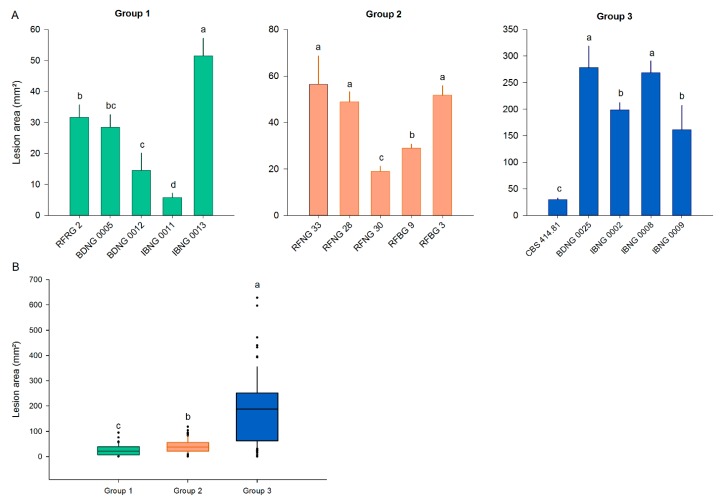
Pathogenicity data of *S. oryzae* isolates of three lineages on Kitaake rice plants. When 7 weeks old, rice plants were inoculated with *S. oryzae* isolates belonging to Groups 1, 2 and 3 using the standard grain inoculum technique. At 8 DPI, the lesion area (mm^2^) was measured. (**A**) Bars show the mean ± SE of each isolate tested, with eighteen replicates for each isolate. Bars marked with different letters are statistically different (Mann–Whitney, *n* = 18, α = 0.05). (**B**) Each boxplot shows the lesion area caused by all five isolates belonging to one lineage. Groups 1, 2 and 3 are represented by, respectively, green, orange and blue boxplots. All boxplots represent the median with the first and third quartile, the whiskers show the minimum and maximum values. Outliers and extreme values are represented by dots. Boxplots marked with different letters are statistically different (Mann–Whitney, *n* = 90, α = 0.05).

**Figure 4 toxins-12-00109-f004:**
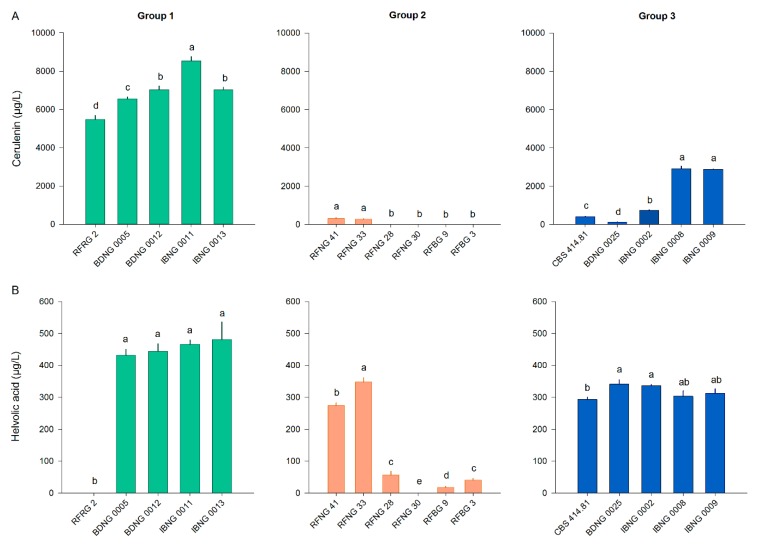
Cerulenin and helvolic acid production by *S. oryzae* isolates on PDA. *S. oryzae* isolates of Groups 1, 2 and 3 were grown for 7 days on PDA. The concentration of (**A**) cerulenin (µg/L) and (**B**) helvolic acid (µg/L) in the extract of the culture was measured with liquid chromatography high-resolution mass spectrometry. Isolates of Groups 1, 2 and 3 are represented by, respectively, green, orange and blue bars. All bars show the mean ± SE of the toxin levels with five replicates for each isolate. Bars marked with different letters are statistically different (Mann–Whitney, *n* = 5, α = 0.05).

**Figure 5 toxins-12-00109-f005:**
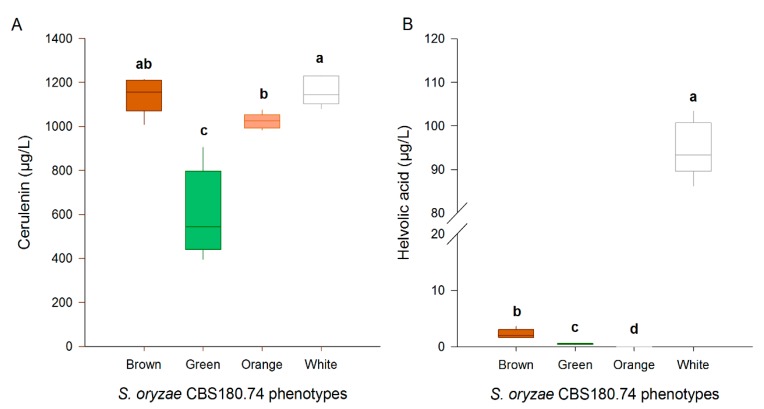
The levels of (**A**) cerulenin and (**B**) helvolic acid produced in vitro by different phenotypes of the *S. oryzae* isolate CBS180.74. Four isolated phenotypes of the Group 1 isolate CBS180.74 were grown in a liquid culture for 7 days. All boxplots represent the median with the first and third quartile; the whiskers show the minimum and maximum values. Boxplots marked with different letters are statistically different (Mann–Whitney, *n* = 5, α = 0.05).

**Figure 6 toxins-12-00109-f006:**
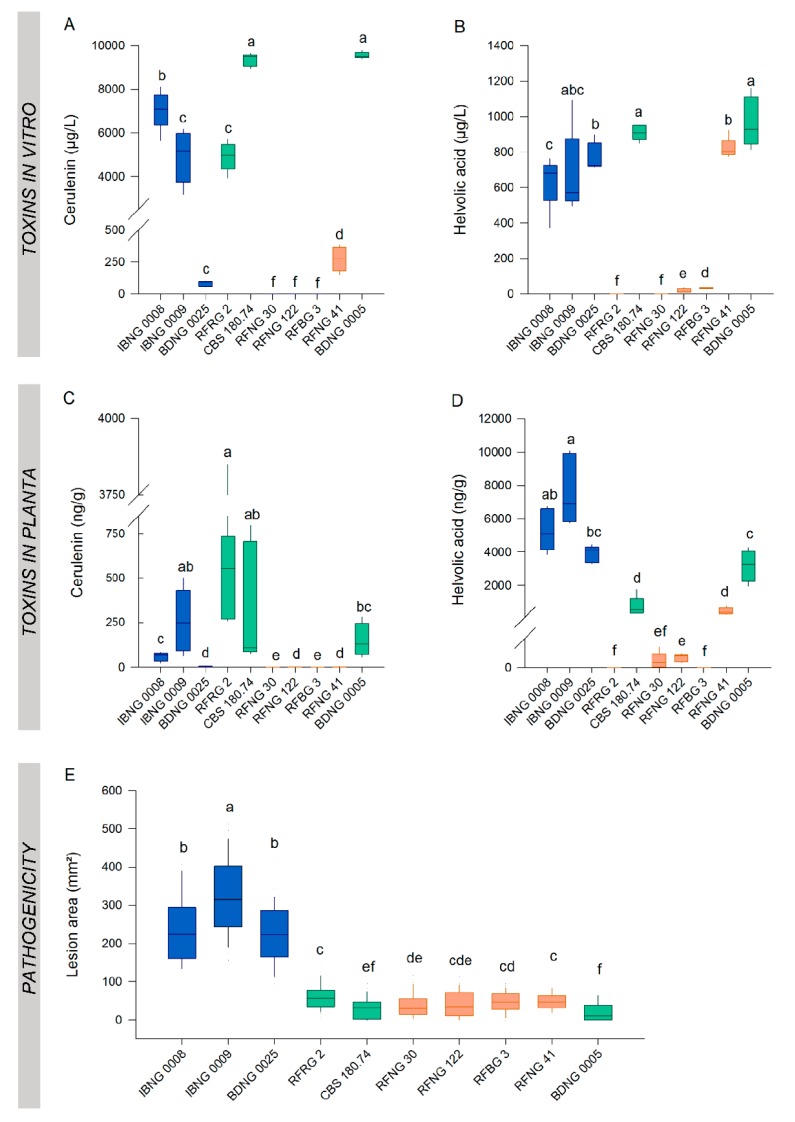
Toxin production in vitro and in planta versus pathogenicity on rice. *S. oryzae* isolates were grown for 7 days on PDA. The concentration of (**A**) cerulenin (µg/L) and (**B**) helvolic acid (µg/L) in the extract of the culture was measured with liquid chromatography high-resolution mass spectrometry. When 7 weeks old, rice plants were inoculated with the same strains of *S. oryzae* using the standard grain inoculum technique. At 6 DPI, (**C**) cerulenin (ng/g) and (**D**) helvolic acid (ng/g) levels were analysed in sheath samples and (**E**) the disease was scored by measuring the lesion area (mm^2^). Isolates of Groups 1, 2 and 3 are represented by, respectively, green, orange and blue boxplots. All boxplots represent the median with the first and third quartile, the whiskers show the minimum and maximum values. Outliers and extreme values are represented by dots. The boxplots show data of 1 experiment with five replicates for each isolate for toxin concentrations (**A–D**) and 25 replicates for each isolate for the lesion area (**E**). Boxplots marked with different letters are statistically different (Mann–Whitney, *n* = 5 or *n* = 25, α = 0.05).

**Figure 7 toxins-12-00109-f007:**
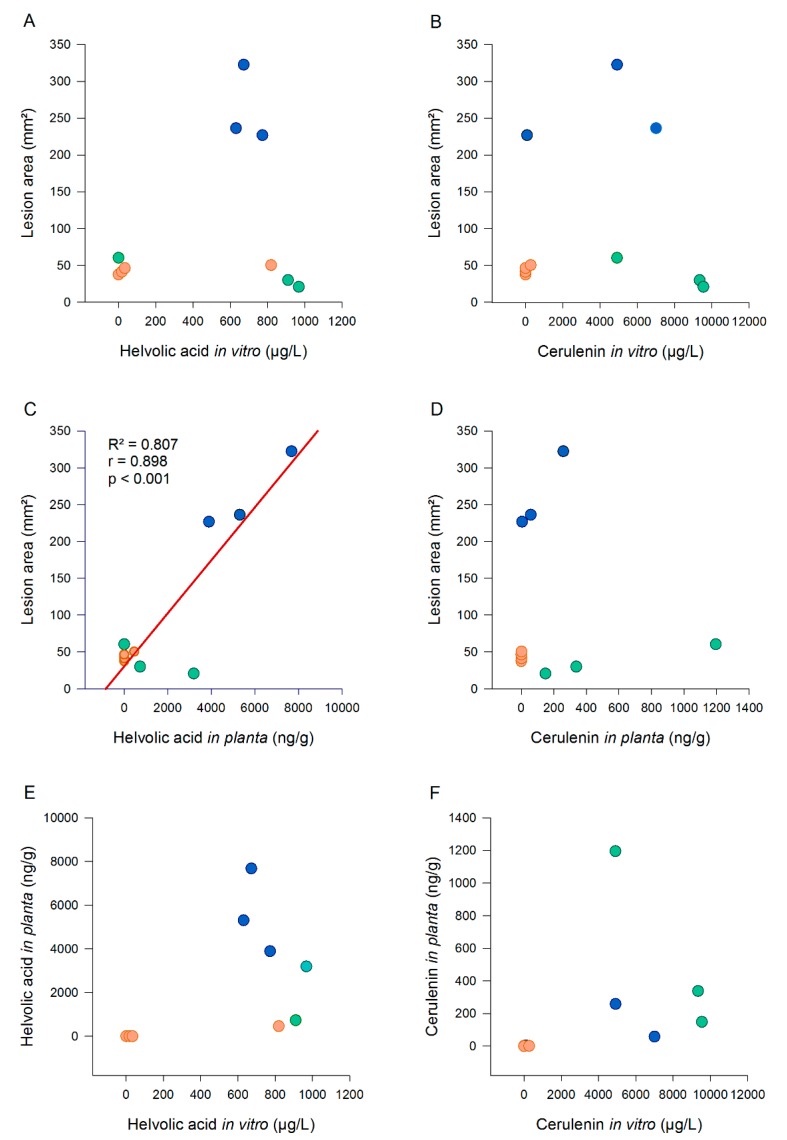
Associations between toxin production in vitro, in planta and pathogenicity on rice. *S. oryzae* isolates were grown for 7 days on PDA. The concentration of cerulenin (µg/L) and helvolic acid (µg/L) in the extract of the culture was measured with liquid chromatography high-resolution mass spectrometry. When 7 weeks old, rice plants were inoculated with the same strains of *S. oryzae* using the standard grain inoculum technique. At 6 DPI, cerulenin (ng/g) and helvolic acid (ng/g) levels were analysed in sheath samples and the disease was scored by measuring the lesion area (mm^2^). Scatterplots show the associations between pathogenicity and helvolic acid and cerulenin levels in vitro (**A**, **B**) and in planta (**C**, **D**) and between the production in vitro and in planta of helvolic acid (**E**) and cerulenin (**F**). Isolates of Groups 1, 2 and 3 are represented by, respectively, green, orange and blue dots; significant linear correlations are represented by the linear regression curve in red.

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
