# Peer review of "Morphological, Pathogenic and Toxigenic Variability in the Rice Sheath Rot Pathogen *Sarocladium Oryzae"

_toxins, 2020, doi:10.3390/toxins12020109_

Round 1
Reviewer 1 Report
Congratulations! This is a very well written article supplemented with good data, statistics, references and neat graphs. I recommend this manuscript for publication.
Pg1 Ln26: 30,000
Pg1 Ln36: You provided a citation, I don't feel you need the sentence "for a review........rot.
Pg2 Ln73: "....specific location of the pathogen" rephrase this sentence
Pg10 Ln181: Start the sentence as "The in vitro production of helvolic acid did not correlate.......
Pg10 Ln183: delete "also"
Pg12 Ln308: ...during 1 h? how about "were extracted in an hour"
Pg13 Ln325: Pathogenicity assays. Is it a subhead?
Pg16 Ln392: BioControl...is it how the Journal cited? Just double check?
Pg17 Ln437: blinded? is it Annonymous?
Pg17 Ln438: Blinded? is it Annonymous?
Pg17 Ln456: Italicize scientific name of Pseudomonas fluorescens
Reviewer 2 Report
Important research work which lays the foundation for plant pathologists to identify resistance mechanisms in rice to rice sheath rot. The research highlighted the importance of helvolic acid as a (likely) key pathogenesis factor.
Value in knocking out the genes involved with the biosynthesis of the antibiotic helvolic acid to confirm the correlation (likely causation) found between strains found to produce high helvolic acid and larger plant lesion areas. Can Sarocladium oryzae be transformed and knockouts produced to test the high producing helvolic strains compared to the low producing strains? Anything referenced in the literature? I see that Professor Monica Höfte (UGent) group is generating Sarocladium oryzae knockouts and perhaps has knockout strains that could be useful using your experimental system.
Reviewer 3 Report
In the submitted manuscript “Morphological, Pathogenic and Toxigenic Variability in the Rice sheath Rot Pathogen Sarocladium oryzae”, the authors characterized the morphology and growth rate of three distinct lineages of fungus Sarocladium oryzae isolated from traditional rice-growing areas in Rwanda and Nigeria, which is a causative agent of sheath rot disease in rice, then they tried to investigate the association of these lineage’s pathogenicity with toxin production by measuring the level of cerulenin and helvolic acid produced by these pathogenic lineages in vitro and in planta using a new high-resolution LC-MS/MS method. They found that the pathogenicity of three lineages was very different on the japonica cultivar Kitaake, there was not correlated between the lineages and helvolic acid production in vitro, but the severity of rice sheath rot disease was strongly associated with the production of helvolic acid in planta, which is a key factor of S. oryzae pathogenicity. The experimental design and data analysis were well done, the conclusions are supported by the data presented in the manuscript.The findings presented in the manuscript will provide a better understand of the role of these toxins during the infectious process of S. oryzae in rice.
Specific point:
I’m wondering whether the authors can describe in more detail of stress tolerance mechanisms, and its relationship with pathogenicity and toxin production in the section of Discussion.
